# Synergistic Effects of DOPO-Based Derivative and Organo-Montmorillonite on Flame Retardancy, Thermal Stability and Mechanical Properties of Polypropylene

**DOI:** 10.3390/polym14122372

**Published:** 2022-06-11

**Authors:** Weijiang Huang, Kui Wang, Chunyun Tu, Xiaolu Xu, Qin Tian, Chao Ma, Qiuping Fu, Wei Yan

**Affiliations:** 1School of Chemistry and Materials Engineering, Guiyang University, Guiyang 550005, China; huangweijiang_2007@126.com (W.H.); gyxywkui@163.com (K.W.); yidapa@sina.cn (C.T.); gyxyxxl@126.com (X.X.); mabtianqin@126.com (Q.T.); chaomagyu@126.com (C.M.); taochao5@163.com (Q.F.); 2National Engineering Research Center for Compounding and Modification of Polymer Materials, Guiyang 550014, China

**Keywords:** polypropylene, DOPO-based derivative, montmorillonite, synergistic effect, flame retardancy

## Abstract

Polypropylene (PP), as a general thermoplastic polymer, is broadly used in different fields. However, the high flammability, melt dripping and poor mechanical properties of PP are a constraint to the expansion of its applications. In this paper, PP composites containing a combination of a phenethyl-bridged DOPO derivative (PN-DOPO) and organic montmorillonite (OMMT) were prepared via melt blending. The synergistic effects of PN-DOPO and OMMT on the flame retardancy, thermal stability and mechanical properties of PP composites were investigated systematically. The results showed that 20 wt% addition of PN-DOPO with OMMT improved the flame retardancy of PP composites. In particular, the introduction of 17 wt% PN-DOPO and 3 wt% OMMT increased the LOI values of the PP matrix from 17.2% to 23.6%, and the sample reached the V-0 level and reduced the heat release rate and total heat release. TGA indicated that OMMT could improve the thermal stability of the PP/PN-DOPO blends and promote the char residues of PP systems. Rheological behaviour showed a higher storage modulus, loss modulus and complex viscosity of PP/PN-DOPO/OMMT composites, suggesting a more effective network structure. In addition, the tensile strength, flexural properties and impact strength of the PP/PN-DOPO/OMMT composites actually increased for a good dispersion effect. Combined with the char layer analysis, the introduction of OMMT promoted more continuous and compact structural layers containing an aluminium–silicon barrier and phosphorus-containing carbonaceous char in the condensed phase. OMMT can improve the flame retardancy, thermal stability and mechanical properties of PP, and, thus, PN-DOPO/OMMT blends can serve as an efficient synergistic system for flame-retarded PP composites.

## 1. Introduction

Polypropylene (PP), a general thermoplastic, is used in automotives, buildings, transportation and electronic devices owing to its excellent electrical insulation properties, corrosion resistance, comprehensive mechanical properties and easy processability [1,2,3]. However, its application is limited by its inherent high flammability, melt dripping and poor impact toughness. This raises the question of how to improve the flame retardancy of PP, improve its toughness and extend its utility [4,5]. Therefore, it is imperative to reduce the fire danger of PP by introducing an environmentally friendly flame retardant and expand its application.

In order to improve the fire resistance of the PP matrix, phosphorus-, silicon- or nitrogen-based flame retardants are usually added into the matrix. The intumescent flame retardants (IFR) composed of charring agents, a gas source and acid sources have attracted interest because of their anti-dripping, low toxicity, low smoke and eco-friendly characteristics [6,7]. Nevertheless, the common IFRs have some apparent disadvantages [8,9,10]. The common IFR systems have poor flame retardancy; more than 30 wt% load in the PP matrix is needed to achieve an ideal flame retardancy [11]. Moreover, its poor compatibility and uneven dispersion in the matrix leads to the deterioration of the comprehensive mechanical properties of the composites. Several studies have attempted to produce flame-retarded PP by adding flame retardants (FRs), such as phosphorus, nitrogen, carbon, silicon-containing FRs, layered double hydroxides, metal hydroxide and transition metal oxides, to minimise the adverse effects of incorporating FRs on the mechanical properties of PP blends [12,13].

Among the commercial halogen-free FRs, phosphorus-based compounds possess unique advantages, such as a high chemical resistance, the nonproduction of toxicants and no transfer. Recently, 9,10-dihydro-9-oxa-10-phosphaphenanthrene-10-oxide (DOPO) and its derivatives have attracted much attention because of their excellent flame-retardant efficiency, high reaction activity and low addition [14,15]. DOPO-based derivatives have been used as novel phosphorus FRs in polymeric materials, such as epoxy resins, polyamides and polyesters [16,17,18]. Synergistic agents usually combine with phosphorus-containing flame retardants to improve the fire resistance of polymers. Generally, the synergistic effect of DOPO-based flame retardants and inorganic particles can reduce the amount of the flame retardant’s additive. The most commonly used synergists mainly involve inorganic particles, such as graphite oxide, sepiolite, montmorillonite, metal oxides and halloysite [19,20,21]. Among them, organic montmorillonite (OMMT) has been proved to be an efficient synergist, which is widely used in polyamide, polypropylene and epoxy resins [22,23,24,25].

OMMT can form a barrier on the surface of a polymer during combustion and can improve the physical and mechanical properties of the polymer matrix at low concentrations. Furthermore, the thermal stability of flame-retarded composites increases with the increasing OMMT amount [26,27,28]. Wen et al. [20] concluded that 20% content of IFR and OMMT had a positive effect and improved the flame retardancy of PP. OMMT could improve the thermal stability of PP composites and promote the residues of PP/IFR blends, but the tensile strength of PP/IFR/OMMT systems was significantly lower than that of pure PP. Du et al. [29] reported flame-retarded PP composites containing IFR and the most popular nanofillers, such as OMMT, layered double hydroxides, polyhedral oligomeric silsesquioxane and carbon nanotubes. The results indicated that the nanofillers could play roles in the thermal and fire resistance of the composites. In particular, PP/OMMT blends gave the best reductions in the heat release rate (HRR) and represented the relatively better flame retardancy. Wang et al. [30] also suggested that the flame retardancy of PP/OMMT/melamine phosphate blends can be enhanced with a low dosage of OMMT. The low OMMT content further reduces the HRR, smoke production rate and mass loss rate. Thus, the OMMT/phosphorus-based FR synergistic system led to higher flame-retardant efficiency and promoted the layer on the surface. However, there are few studies on the synergistic flame-retardant effects of an OMMT and DOPO-based system for polypropylene.

Based on our previous reports, a novel phenethyl-bridged DOPO derivative (PN-DOPO) was prepared. The effects of PN-DOPO on the flame-retardant and thermal properties of the polymeric matrix, such as polyamide, epoxy resins and polylactide, were examined [17,31,32,33,34,35]. The results showed that PN-DOPO could be a good candidate to improve the flame retardancy and thermal stability of the polymer matrix. According to the research reports, the combination of the DOPO-based derivative and nanoparticles could also effectively improve fire resistance and the thermal stability of PP [4,5]. We can design it so that PN-DOPO is combined with inorganic particles to obtain higher flame-retardant efficiency. Hence, this study aimed to improve the flame retardancy, thermal stability and mechanical properties of PP using a combination of PN-DOPO and OMMT.

In this study, PP/PN-DOPO/OMMT composites were prepared through melt mixing. The synergistic flame-retarded effect, thermal stability and rheological behaviour of PN-DOPO combined with OMMT on PP was evaluated. The composition structure and morphology of char residues were investigated systematically. Furthermore, the possible synergistic effect of PN-DOPO and OMMT on the flammability and mechanical properties of flame-retarded PP were also discussed.

## 2. Materials and Methods

### 2.1. Materials

Polypropylene resin (F401) was purchased from Yangzi Petroleum Chemical Company (Nanjing, China) with a melt flow index of 2.5 g/min (230 °C, 2.16 kg). The FR, phenethyl-bridged DOPO derivatives (PN-DOPO, were synthesised in our laboratory according to the previous reports [34,36]. The chemical structure is shown in Figure 1. The OMMT (I.30 P), was purchased from Nanocor Inc. (Arlington Heights, Cook County, Illinois, USA).

### 2.2. Preparation of Flame-Retarded PP Composites

PP, PN-DOPO and OMMT were dried at 80 °C under vacuum for 10 h before being used. PP/PN-DOPO and PP/PN-DOPO/OMMT composites were prepared by melt blending in a torque rheometer (Haake PolyLab OS, Thermo Fisher Scientific Co., Ltd., Karlsruhe, Germany) at 190 °C with a screw speed of 50 rpm for 6 min. Table 1 lists the composition of the formulation. Subsequently, the mixtures were hot-pressed first at 190 °C for 10 min at 15 MPa. They were then cold-pressed at room temperature for 20 min with 15 MPa into sheets of suitable size and thickness for flame retardancy and mechanical properties tests using a plate vulcaniser (ZHY-W, Chengde Testing Machine Factory, Hebei, China).

### 2.3. Characterization

The UL-94 vertical combustion test was performed according to ASTM D3801 with a sample size of 130.0 × 13.0 × 3.2 mm. The LOI values were obtained using the LOI instrument according to ASTM D2863-06. The sample dimensions were 130.0 × 6.5 × 3.2 mm. Microscale combustion calorimeter (MCC) tests were conducted using an MCC-2 instrument (Govmark Co., Ltd., Farmingdale, NY, USA). The powder samples (approximately 5–7 mg) were heated from 40 °C to 650 °C at a heating rate of 1 °C/s in an aluminium crucible. The heating atmosphere is a mixture of nitrogen flowing at 80 mL/min and oxygen flowing at 20 mL/min.

The thermal stability of the samples was analysed via thermogravimetric analysis (TGA) (TG 219 F3, Netzsch Instruments Co., Ltd., Selbu, Germany) under nitrogen atmosphere at a flow rate of 60 mL/min. Each sample (approximately 6–10 mg) was heated from 30 °C to 700 °C at 10 °C/min heating rate.

The morphology of the char residues after UL-94 tests was obtained via SEM (Quanta FEG 250, FEI Instruments, Hillsborough, Oregon, USA) at an acceleration voltage of 10 kV.

Energy dispersive spectrometry (EDS) (INCA-350, Oxford, UK) was performed to characterise the chemical composition of the residues. EDS analyses were recorded on SEM equipped with an EDS accessory under high vacuum conditions at a voltage of 30 kV.

X-ray photoelectron spectroscopy (XPS) (Escalab Mark II, VG Scientific Ltd., St. Leonards-On-Sea, UK) was performed using an Al Kα source at a base pressure of 1.0 × 10^−8^ mbar. The C, O, P, Si and Al on the char residue of the samples were investigated.

The rheological behaviour of samples was analysed using a Rheometric Analyzer (HAAKE MARSII, Thermo Fisher Scientific Inc., Karlsruhe, Germany) using parallel plates with diameters of 35 mm. Dynamic frequency sweep tests were measured at 190 °C using 1% strain in an angular frequency range of 0.1–100 rad/s.

The mechanical properties of flame-retarded PP composites were measured using a CMT6104 universal testing machine (MTS Systems Co., Ltd., Shanghai, China). The tensile tests were measured according to ASTM D-638 using an extensometer. The flexural tests were performed according to ASTM D-790. For each sample, the results were an average of five specimens, and the range of standard deviations was within 5%. A notched Izod impact test was experimented using a pendulum impact tester (ZBC-4B, Sans Co., Shenzhen, China) according to ASTM D-256.

## 3. Results and Discussion

### 3.1. Flame Retardancy of PP Composites

The UL-94 test and LOI measurement are widely used to evaluate the flame retardancy of polymers. Table 1 lists the corresponding experimental results, and Figure 1 shows digital images of PP composites after the UL-94 test. The pure PP sample had a LOI value of only 17.2% with heavy dripping, and no significant char formation was observed during the tests. The recorded flame times were more than 60 s, and no rating was reached. As shown in Table 1, t_1_ and t_2_ in the UL-94 test decreased gradually with increasing PN-DOPO content. Dripping was no longer observed when the PN-DOPO content was 15 wt%, and a UL-94 V-1 rating was attained. For PP/20 wt% PN-DOPO, the LOI value increased to 23.1% and attained a UL-94 test V-0 rating. The improvement in the flame retardancy of PP/PN-DOPO composites was apparent in the UL94 and LOI flammability test. This behaviour is mainly attributable to the principal gas phase activity of the DOPO derivatives [37].

**Table 1 polymers-14-02372-t001:** Composition of the PP composites and results of UL-94 vertical combustion and LOI tests.

Samples	PP (wt%)	PN-DOPO (wt%)	OMMT (wt%)	LOI (%)	UL-94 (3.2 mm)		
					t_1_/t_2_ (s)	Dripping	Rating
PP	100	0	0	17.2	>60/--	Yes	No Rating
PP-1	90	10	0	21.7	8.1/16.7	Yes	V-2
PP-2	85	15	0	22.5	7.5/11.2	No	V-1
PP-3	80	20	0	23.1	5.2/2.6	No	V-0
PP-4	75	25	0	23.7	2.9/1.5	No	V-0
PP-5	80	19	1	23.4	7.8/0.5	No	V-0
PP-6	80	17	3	23.6	8.3/0.8	No	V-0
PP-7	80	15	5	24.0	15.2/0.7	No	V-1
PP-8	80	13	7	23.8	21.6/2.9	No	V-1
PP-9	93	0	7	20.3	>60/--	No	No Rating

OMMT was incorporated into the PP/PN-DOPO composites to improve the flame-retardancy efficiency further and decrease the need for high loadings (20 wt%). The loading of PN-DOPO/PMMT compounds was maintained at 20 wt% in the PP samples. When 1 wt% and 3 wt% OMMT replaced the same content of PN-DOPO, respectively, the V-0 rating was reached, and the variation in LOI values was not evident. With increases in the OMMT amount, the UL-94 rating of V-1 was achieved for both PP-7 and PP-8. Moreover, the LOI values were slightly increased to 24.0% and 23.8%, respectively, which were higher than that of PP-3. However, for only 7 wt% OMMT content, the LOI value of PP-9 was 20.3%, and no rating was reached.

As shown in Figure 1, upon the ignition of neat PP, it burnt out without leaving any char residue. The flame rapidly propagated after ignition, accompanied with vigorous dripping. Compared with pure PP, the combustion behaviours of PP/20 wt% PN-DOPO and PP/PN-DOPO/OMMT composites were significantly different, and combustion dripping was not observed. Figure 1 shows that almost no char was formed from PP/PN-DOPO after the UL-94 test. In contrast, OMMT-containing PP composites formed visible chars after burning. This may be due to its layered carbonated structure. During the combustion, the addition of OMMT increased the melt viscosity of the matrix, and improved the char formation ability and anti-dripping properties to some extent [22,38]. Flammability test results indicated that the addition of PN-DOPO and the modest amount of OMMT had a synergistic flame-retardant effect, and the combination could further improve the flame retardancy of the PP matrix.

The MCC test is an effective method to investigate the combustion properties of polymers. The combustion behaviour of PP composites was analysed using MCC tests. Some important parameters, such as the specific HRR, total heat release (THR) and temperature of peak HRR (T_pHRR_), could be obtained from MCC tests [21,39]. The corresponding data are summarized in Table 2. Figure 2 presents the HRR curves of PP and flame-retarded PP composites.

Figure 2 shows that the neat PP had the highest pHRR of 1260.1 W/g. With the addition of 20 wt% PN-DOPO, the pHRR value decreased to 1162.3 W/g. With the addition of OMMT into the PP/PN-DOPO blends, the pHRR values of composites were further reduced. When the amount of OMMT was 1 wt% and 3 wt%, the pHRR values of PP-5 and PP-6 decreased to 1037.7 W/g and 914.0 W/g, respectively. The pHRR values were reduced by 17.6% and 27.5% compared with PP, respectively. However, when the OMMT loading increased from 5 wt% to 7 wt% OMMT, the pHRR values increased from 976.4 W/g to 1118.3 W/g, respectively. This demonstrated that 3 wt% content of OMMT into PP/PN-DOPO could reduce the pHRR value to the greatest extent. The results accorded well with the UL-94 vertical combustion test results.

THR is another important parameter for assessing fire disasters, which is calculated according to the area under the HRR curve. As listed in Table 2, the neat PP had the highest THR value of 49.1 kJ/g. In comparison, all PP composites with OMMT had a lower THR value. Moreover, PP/17 wt% PN-DOPO/3 wt% OMMT showed the lowest THR with a value of 42.7 kJ/g among these PP composites, and the THR value was decreased by 13.0% compared with that of PP.

As shown in Table 2, the T_pHRR_ of PP was 468.6 °C. For PP/20 wt% PN-DOPO, the T_pHRR_ of PP-3 was 467.9 °C. After introducing 1–7 wt% OMMT, the T_pHRR_ value remained within 477.2–485.1 °C. The increase in T_pHRR_ value indicated that PP/PN-DOPO/OMMT composites were more difficult to burn than PP and PP/PN-DOPO and could delay the decomposition process of the matrix [39], which enhances the flame retardancy of PP.

From the analyses above, OMMT was efficient in depressing the heat release of PP combustion. The combination of PN-DOPO and OMMT exhibited obvious synergistic flame retardancy.

### 3.2. Thermal Stability

Figure 3 shows the TGA and DTG curves of the samples, and Table 3 lists the relevant results. The corresponding data included T_5%_, which is defined as the initial decomposition temperature at 5 wt% weight loss; T_max_, which is defined as the temperature at the maximum weight loss rate; and the final char residues at 700 °C.

TGA and DTG curves (Figure 3) showed that the thermal decomposition of neat PP occurred in a single-step weight loss in the temperature range of 390–490 °C, and T_5%_ and T_max_ were 410.7 °C and 455.3 °C, respectively. After the temperature was increased gradually, weight loss increased rapidly, leaving negligible char residue at 700 °C. With the addition of 20 wt% PN-DOPO, the decomposition of PP/PN-DOPO blends also showed a one-step mass loss, with the T_5%_ and T_max_ at 367.8 °C and 459.2 °C, respectively. The final char residue was only 0.16 wt%. The addition of DOPO-based FRs reduce the initial thermal stability of the polymer matrix and slightly promotes char formation [17,18]. These results are in agreement with other similar reported conclusions [22,31,35]. After adding OMMT into PP, the peak temperatures of mass loss were reduced to 432.7 °C, which was mainly due to the decomposition of organoclay [20,40]. After OMMT was combined with PN-DOPO into PP, the thermal decomposition behaviours of the composites were changed markedly compared with the PP/PN-DOPO sample. The apparent increases in T_5%_ and T_max_ reveal the improvement in thermal stability of the PP matrix with the introduction of OMMT. Remarkably, the residues’ amount of the PP/PN-DOPO/OMMT composites increased with the increasing combination. The reason may be ascribed to the effect of OMMT on the thermal decomposition process of PP or the different interaction between PN-DOPO/OMMT and the matrix [41,42]. When the amount of OMMT was 3 wt%, the sample had the highest T_5%_ and T_max_ of 378.3 °C and 468.2 °C, respectively. Compared with the PP/PN-DOPO sample, the T_5%_ and T_max_ increased by 10.5 °C and 9.0 °C, respectively, and the residues increased to 2.2 wt%. The reason for the increment in thermal stability and char residue can be attributed to a synergistic effect on the formation of protective barrier layers or the thermal interaction between the flame retardants and the polymer matrix [31,43].

### 3.3. Morphology of the Char Residue

SEM was performed on the residues after the UL-94 tests. Figure 4 presents the microscopic morphologies of the char residues for PP-3, PP-6, PP-8 and PP-9. Pure PP had no char residue left after burning (see Figure 1); hence, no SEM images of PP were recorded. As displayed in Figure 4a, broken holes, flaws and many voids could be observed on the surface of the char for the PP/PN-DOPO sample [5,43]. By comparison, after adding OMMT into the PP/PN-DOPO blends, the charred layers of composites showed more continuous and compact structural layers without cracks, which would act as a heat insulation barrier. The structures could impede the permeability of the heat and gases and enhance the flame retardancy of the composites. Similar results were reported by Yang et al. for the EP/DOPO/MMT [44] system and He et al. for the PA6/ABPA/MMT [40] system. Based on the morphology of the residue, PP/PN-DOPO/OMMT composites have an apparent synergistic effect on the char formation.

### 3.4. Chemical Structure of the Char Residues

The element compositions of the char residues were characterised by EDS, and Table 4 lists the corresponding results. For the PP/20%PN-DOPO sample, high O, P and C contents in the char residue were detected. The residue was composed of aromatic structures and phosphorus oxide compounds [31,45,46]. The residue of PP/7%OMMT composites consisted of high O, Si, Al and Mg contents and relatively low carbon contents. According to Ronkay et al. [47] and Bourbigot et al. [48], these elements were the main components of layered montmorillonite. After adding OMMT into the PP/PN-DOPO blends, an increase in the carbon and OMMT components (silicon, aluminium and magnesium) was observed. By comparing the EDS data of the samples, the C, Si, Al and Mg mass ratios of the PP/PN-DOPO/OMMT residues were higher. The results show that the large C content is locked in the condensed phase. The high C/O ratio can usually improve the anti-oxidation ability and crosslinking density of the char residues [31,49].

XPS can be used to examine the chemical structure of char residues. The composition of residues was also characterised by XPS. Figure 5 and Figure 6 present the XPS spectra of the residues for the flame-retarded PP composites and Table 5 lists the corresponding results. As shown in Figure 5, the C_1s_, O_1s_, P_2p_, Si_2p_ and Al_2p_ peaks of PP/17%PN-DOPO/3%OMMT were observed; the spectra exhibited an increased intensity at the C_1s_ peak compared with PP/20%PN-DOPO and PP/7%OMMT. The residue of PP/20%PN-DOPO contained mainly C, O and P. From the spectra of PP/20%PN-DOPO, the C_1s_ spectra displayed two bands with binding energies of 284.6 and 286.3 eV, respectively. The 284.6 eV band was ascribed to the C–H or C–C bonds in aliphatic and aromatic structures. The C_1s_ peak at 286.3 eV was attributed to the C–O group. The O_1s_ spectra featured two bands with 532.2 and 533.7 eV binding energies, which were assigned to –O– groups in P–O–C bonds and C=O groups in phosphate or carbonyl compounds, respectively [11,50]. The bands of P_2p_ (133.1 and 134.6 eV) originated mainly from the P–O and P=O bonds. From the spectrum of PP/7%OMMT, the C_1s_ spectra featured two bands with binding energies of 284.6 and 285.7 eV. For O_1s_ spectra, the 532.3 eV peak can be attributed to the –O– bond. The 533.6 eV binding energy was ascribed to the C=O groups in carbonyl compounds. A single peak at 102.8 eV in the Si_2p_ spectrum was assigned to the Si–O bond in clay platelets. The Si–O–Si structure generally acts as an enhanced framework for char formation and provides a synergistic effect on flame-retarded composites [21].

From the spectrum of PP/17%PN-DOPO/3%OMMT, the C_1s_ spectra have three bands with binding energies of 284.7, 285.7 and 288.1 eV. The 284.7 and 285.7 eV peaks were assigned to the C–H or C–C groups and C–O group, respectively. Furthermore, the binding energies of 288.1 eV can be attributed to the C=O or carbonyl groups [51]. Hence, the C content increased from 73.3 at% for PP/20%PN-DOPO to 80.7 at% for PP/17%PN-DOPO/3%OMMT, indicating that the PN-DOPO/OMMT system can participate in and catalyse char formation. The aromatic carbons (C=C-C group) and phosphaphenanthrene contents decomposed by PN-DOPO [17] are crosslinked with the aluminium–silicate barrier (Al–O and Si–O structure from OMMT) to form a more continuous and compact layer. The change in the C, O, P, Al and Si contents in the residual char was similar to the EDS results.

Overall, XPS confirmed that the hybrid of phosphorus-, silicon- and aluminium-containing compounds could promote a more continuous and compact char layer. The char layer can effectively decelerate the mass and heat transfer between the solid and gas phases. In this case, the PP/PN-DOPO/OMMT composite exhibited a synergistic flame-retardant effect on char formation.

### 3.5. Rheological Behaviour

The influence of the melt flow characteristics on the combustion behaviour of flame-retarded composites’ nanoclay-containing and phosphorus-containing FRs was investigated [40,52]. The structural morphology of the dispersed phase, melt flow of composites and blend ratio affect the rheological behaviour of composites [53,54,55], and the rheology is a powerful method for analysing the internal structure of composites. The effects of OMMT/PN-DOPO compounds on the PP matrix were examined. Figure 7 and Figure 8 show the variation in the storage modulus (G’), loss modulus (G’’) and complex viscosity (η*) as a function of the frequency for PP and flame-retarded PP composites at 190 °C, respectively. The G’ and G’’ values of the flame-retarded PP composites increased compared to the PP values over the entire frequency range. In the high ω regime (>10 rad/s), all samples showed the same trend, and there were no obvious differences in the slopes of the curves (Figure 7). The G’ and G’’ of PP/PN-DOPO/OMMT composites were significantly higher than those of PP and PP/OMMT with the introduction of PN-DOPO and OMMT in the low frequency range (<0.1 rad/s). The G’ and G’’ values increased with increasing OMMT content in the blends. The slope of log G’ versus the log ω plot was also extremely small in the range. The terminal behaviour disappeared at the low frequency when PN-DOPO and OMMT were added to the matrix, indicating a transition from liquid-like to solid-like viscoelastic behaviour for flame-retarded composites [52]. Similar solid-like viscoelastic behaviour responses and intracycle strain thickening results from the formation of a percolation network in composites owing to the confinement effect of nanofillers on the motion of the polymer chain have been reported previously [53,56].

Similar trends are observed in the complex viscosity of PP and flame-retarded PP composites (Figure 8). The viscosity increased with the increasing OMMT amount for PP/PN-DOPO/OMMT composites over the entire frequency range. The η* values of PP/PN-DOPO/OMMT composites were significantly higher than the values of PP and PP/OMMT in the low frequency range. The PP/13%PN-DOPO/7%OMMT composites showed the highest η* value. Hence, a stronger network structure was built in the PP/PN-DOPO/OMMT composites because the variations in the dynamic viscosity and modulus in the low frequency range were characteristic of network-like structures [40]. The network structure can enhance the melt viscosity and limit the fluidity of polymer chains during burning. Furthermore, the clay network also increased the barrier characteristics of oxygen entering the condensed phase and the evolution of combustible volatiles, resulting in improvements in the flame retardancy of composites [57].

### 3.6. Synergistic Flame-Retardant Mechanism

Based on the above results, the possible synergistic flame-retardant effect between OMMT and PN-DOPO can be discussed as follows. For pure PP, it burnt out without leaving any char residue, accompanied with vigorous dripping during combustion. For PP/PN-DOPO blends, the char residue of the sample with broken holes and many voids in the cavity structure was observed owing to gas phase quenching at the initial stage of combustion [17,31]. With the incorporation of OMMT into PP/PN-DOPO blends, the surface of the PP/PN-DOPO/OMMT composites showed a continuous and compact structural layer after burning. The aromatic carbons (C=C-C group) and phosphaphenanthrene contents decomposed by PN-DOPO are crosslinked with the aluminium–silicate barrier (Al–O and Si–O structure from OMMT) to form a more compact layer. Furthermore, the physical crosslinking network structure can enhance the melt viscosity and limit the fluidity of polymer chains during burning. The charred residue prevented the transportation of degradation products and heat transfer, thus resulting in higher flame-retardant efficiency and thermal stability. Hence, the incorporation of OMMT can effectively form a dense and continuous charred layer and significantly improve the flame retardancy and thermal stability of the matrix, and exhibit an apparent synergistic flame-retardant effect.

### 3.7. Mechanical Properties of PP Composites

The mechanical properties were investigated to evaluate the effects of OMMT/PN-DOPO compounds on PP. Figure 9 shows the mechanical properties and stress–strain curves; Table 6 lists the corresponding results. Neat PP showed high elongation at break but low flexural properties and tensile strength. With a 20 wt% PN-DOPO loading, the tensile strength, flexural strength and modulus increased by 14.1%, 21.0% and 23.7%, respectively, whereas the elongation at break and notched impact strength decreased slightly. For PP/7%OMMT, the addition of OMMT alone resulted in a significant decrease in the mechanical properties of the sample. For PP/PN-DOPO/OMMT blends, the tensile strength, flexural properties and impact strength improved gradually with increasing OMMT content. With the incorporation of 5 wt% OMMT, the mechanical property of PP-7 was the highest among the PP/PN-DOPO/OMMT systems. For PP-7, the tensile strength, flexural strength, flexural modulus and impact strength were increased by 37.3%, 34.2%, 68.8% and 36.4%, respectively, compared with neat PP. The enhanced tensile strength, impact strength and flexural properties of PP composites may be due to the fact that PN-DOPO and OMMT could be uniformly dispersed in the matrix during the hot melt processing. These results were consistent with the conclusions reported in the previous literature [58,59]. Furthermore, the elongation at break decreased with increasing OMMT content (as shown in Figure 9b). Although no enhancement in nominal elongation at break was observed after addition, the increased impact strength of the PP/PN-DOPO/OMMT composites indicated that OMMT/PN-DOPO compounds were efficient for maintaining the toughness of the matrix.

The mechanical properties of composites depend on the compatibility of the fillers within the matrix. Figure 10 shows the SEM image of the impact fracture of the sample. The surface of neat PP was relatively smooth and flat, which was typical of a brittle failure feature. In contrast to the gently undulating surface of PP, there were some dents and bumps dispersed unevenly on the PP/20%PN-DOPO surface (Figure 10(b1,b2)). Compared with neat PP and PP/20%PN-DOPO, the fracture surface of the flame-retarded PP/PN-DOPO/OMMT composites (PP-7) were rougher and stratified. More uniform and fine dispersed fillers are observed in Figure 10(c1–c3). There is no obvious phenomenon of agglomerates within PP matrices, which indicates a good compatibility between PN-DOPO/OMMT and the matrix. This may be due to the exfoliation of OMMT during the melting process, resulting in better dispersion of PN-DOPO/OMMT particles in the matrix [59].

This OMMT layer and flame retardant played a reinforcing role in PP composites. Additionally, the tensile strength and flexural property of composites were improved [58] with the increasing proportion of OMMT in PP/PN-DOPO/OMMT blends. Hence, PP-7 composites presented the greatest tensile strength, flexural property and impact strength.

In conclusion, when OMMT and PN-DOPO were applied to flame-retarded PP, the composites imparted a presented significant synergistic flame-retardant effect on the combustion behaviour, melt viscosity, thermal stability and char formation. In addition, they also enhanced the mechanical properties, such as tensile strength, flexural properties and impact strength, compared with the PP/PN-DOPO system.

## 4. Conclusions

Different ratios of PN-DOPO combined with OMMT were blended into PP to improve the flame retardancy of the samples. The effects of PN-DOPO/OMMT compounds on the flame retardancy, thermal stability and mechanical properties of the composites and the char morphology of the samples after burning were investigated systematically. When adding 17%PN-DOPO and 3%OMMT, the LOI value for PP/PN-DOPO/ OMMT composites increased to 23.6%, and the UL-94 test reached the V-0 level. The pHRR and THR values of the sample were reduced by 27.5% and 13.0%, respectively, compared with those of PP. TGA showed that the combination of PN-DOPO and OMMT could improve the thermal stability and the content of residues of the PP systems. PN-DOPO/OMMT blends could promote a more continuous and compact structural layer with a high yield of the aluminium–silicate barrier and phosphorus-containing carbonaceous compounds. Furthermore, the PP/PN-DOPO/OMMT composites showed higher tensile strength, flexural properties and impact strength than those of the GFPA6T/PN-DOPO mixtures. In summary, OMMT can be a highly effective and promising synergistic agent to improve the flame retardancy and mechanical properties of flame-retarded PP composites.

## Data Availability

The data presented in this study are available on request from the corresponding author.

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
