# Peer review of "Synergistic Effects of DOPO-Based Derivative and Organo-Montmorillonite on Flame Retardancy, Thermal Stability and Mechanical Properties of Polypropylene"

_polymers, 2022, doi:10.3390/polym14122372_

Round 1
Reviewer 1 Report
The manuscript titled; “Synergistic effects of DOPO-based derivative and organo-montmorillonite on flame retardancy, thermal stability and mechanical properties of polypropylene” provides details on PP processing and its optimizations. Some comments are suggested below;
1. The carried research is interesting but the writing should be improved.
2. The literature review on polymer processing and rheological behaviors is too little, other processes parameters should be discussed, such blend ratio, melt flow and crystallinity. Followings can be added in terms of variant polymer processing dynamics:
a. Viscoelastic Rheological Behaviors of Polypropylene and LMPP Blends
b. LMPP effects on morphology, crystallization, thermal and mechanical properties of iPP/LMPP blend fibres
c. Optimization of mechanical and thermal properties of iPP and LMPP blend fibres by surface response methodology
3. Can authors add comments on the interactions of flame retardants and PP polymer.
4. The mechanical attributes should be connected with the PP processing and flame retardant in it.
5. The conclusion part should be concise to the main findings only. Good work.
Author Response
Dear editors and reviewers:
Thank you for your letter and for the reviewers’ comments concerning our manuscript entitled “Synergistic effects of DOPO-based derivative and organo-montmorillonite on flame retardancy, thermal stability and mechanical properties of polypropylene” (Manuscript ID: polymers-1749651).
Those comments are very helpful for revising and improving our paper, as well as the important guiding significance to our researches. We have studied the all comments carefully and have made correction which we hope to meet with approval, and the revised portion are marked in red in revised manuscript. The main corrections in the revised paper and the detailed responses to the reviewers one by one are listed as follows.

Reviewer 2 Report
The manuscript 'Synergistic effects of DOPO-based derivative and organo-montmorillonite on flame retardancy, thermal stability and mechanical properties of polypropylene' describes the synthesis and characterization of PP-based composites with 9,10-dihydro-9-oxa-10-phosphaphenanthrene-10-oxide (DOPO) and inorganic filler. The results of the study are very promising and presented perfectly, and I recommend only minor revision to improve the quality of presentation.
My recommendations:
According to Polymers template, the Abstract should contain short introductory part (brief description and actuality of the problem). Please keep in mind during preparation of the revised manuscript
Line 13 – organo-montmorillonite
2. Materials and Methods
Line 122 and below – please follow the rules of citation, do not use superscript
Author Response
Dear editors and reviewers:
Thank you for your letter and for the reviewers’ comments concerning our manuscript entitled “Synergistic effects of DOPO-based derivative and organo-montmorillonite on flame retardancy, thermal stability and mechanical properties of polypropylene” (Manuscript ID: polymers-1749651).
Those comments are very helpful for revising and improving our paper, as well as the important guiding significance to our researches. We have studied the all comments carefully and have made correction which we hope to meet with approval, and the revised portion are marked in red in revised manuscript. The main corrections in the revised paper and the detailed responses to the reviewers one by one are listed.

Reviewer 3 Report
In the reviewed work a 9,10-dihydro-9-oxa-10-phosphaphenanthrene- 10-oxide (DOPO) phenethyl-bridged derivative (PN-DOPO) and organo-montmorillonite (OMMT) were melt blended with polypropylene (PP), and the resulting composites were characterized by TGA, SEM and XPS. Flame retardancy, thermal stability and mechanical properties were evaluated and it has been found that N-DOPO increased the LOI values of PP matrix. The research idea presented in this work is sound, however, there are several points that need to be addressed in more depth:
- please provide the chemical structure of PN-DOPO,
- please provide TG profiles of )-MMT and PN-DOPO to evidence that these additives do not undergo the thermal decompositon at processing temperature of 190 deg C,
- nanofillers, such as MMT, tend to form agglomerates within polymer matrices – please show HR-SEM images of PP –based (nano)composites,
- what is the compatibility of PN-DOPO with PP – it was compatible polyamide, epoxy resins and polylactide (Introduction: Lines 99/100), however, they are polar polymer
- was O-MMT intercalated or exfoliated in PP matrix? Please provide WAXD data and calculate the average spacing using Scherrer equstion,
- please discuss in more detail the mechanism of PN-DOPO / O-MMT flame ratardation synergistic action,
- Conclusions: „The morphology and chemical structure of the char residues confirmed that a high yield of aluminium-silicate barrier and phosphorus-containing carbonaceous char contributed to the formation of more continuous and compact structural layers…” – however, TGA (Fig. 3) results do not show char formation, do they?
Author Response

(The authors gave the same response as above.)

Round 2
Reviewer 1 Report
No comments to add.
Reviewer 3 Report
Authors provided proper explanation to my comments & questions, and the revised manuscript can be published as it is.